# Developments in Leishmaniasis diagnosis: A patent landscape from 2010 to 2022

**Daniel Moreira de Avelar[1], Camila Chaves Santos[2], Alice Fusaro Faioli**[1,3]*

**1** Instituto René Rachou—Fiocruz Minas, Belo Horizonte-MG, Brazil, **2** Instituto Nacional de Propriedade Intelectual–INPI, Rio de Janeiro-RJ, Brazil, **3** Fundação Oswaldo Cruz-Fiocruz, Centro de Desenvolvimento Tecnológico em Saúde, Instituto Nacional de Ciência e Tecnologia de Inovação em Doenças de Populações Negligenciadas, Rio de Janeiro, RJ, Brasil

* alice.faioli@fiocruz.br

## Abstract

The current study aims to contribute to the understanding of leishmaniasis diagnosis by providing an overview of patent filings in this field and analyzing whether the methods revealed are consistent with the needs described by the scientific community, in special the main gaps detected by the World Health Organization's 2021–2030 Roadmap for Neglected Tropical Diseases. To this aim, a patent search was carried out focusing on documents disclosing leishmaniasis diagnostic methods supported by experimental evidence and with earliest priority date from 2010 onwards. Our results show that patenting activity is low and patent families are often formed by individual filings. Most R&D activity occurs in Brazil, which is also the main market of protection. Brazilian academic institutions are the main patent drivers, and collaboration between different institutions is rare. Most patent families describe immunological methods based on ELISA assays, using antibodies directed to K39 and homologues. *kDNA* is the primary gene for molecular testing. Experimental evidence of test performance in fulfilling critical diagnostic gaps is usually absent. The patent scenario suggests that leishmaniasis diagnostic gaps need to be more closely addressed to drive innovation directed to the control and/or elimination of leishmaniasis. From the public policy point of view, the following strategies are suggested: (i) strengthening collaborative networks, (ii) enhancing the participation of the private sector, and (iii) increasing funding, with special focus on the remaining diagnostic gaps.

## Introduction

Leishmaniasis is an important disease complex caused by protozoan parasites of the genus *Leishmania*. It is usually classified according to clinical presentation as tegumentary (TL) or visceral (VL) leishmaniasis. Cutaneous (CL) and mucocutaneous (MCL) leishmaniasis are clinical forms of TL, while Post Kala-azar Dermal Leishmaniasis (PKDL) is a late cutaneous manifestation of VL. Leishmaniasis is considered a neglected tropical disease (NTD), with an estimated 700,000 to 1 million cases annually.

Significant efforts have been made in the past decades to control and/or eliminate NTDs, leishmaniasis included. Among key global actions are stakeholder commitments such as the

**Data Availability Statement:** The dataset used for this work is provided as supplementary material.

**Funding:** This work was funded by FAPEMIG via the following grants: ACN-00110-21 (AFF), and APQ-00802-20 (DMA). Funding for publication was

provided by CNPq, CAPES, and FAPERJ through the National Institutes of Science and Technology Program (INCT) to Carlos Morel (INCT/IDPN). The funders had no role in study design, data collection and analysis, decision to publish, or preparation of the manuscript.

**Competing interests:** The authors have declared that no competing interests exist.

London Declaration and the 2030 Agenda for Sustainable Development [1, 2], and WHO roadmaps for NTDs (the first encompassing the years of 2012 to 2020 and the second from 2021 to 2030) [3, 4].

Specific goals for leishmaniasis set in WHO's 2021–2030 roadmap are: eliminating VL as a public health problem in 85% of countries and controlling CL (85% of all cases detected and reported, and 95% of reported cases treated). The roadmap recognizes diagnostics play a central role towards the achievement of such goals [4]. Effective diagnostics accelerate case detection and treatment, reducing disease progression and ensuing disability, and contributing to disease eradication by reducing sources of infection. In addition, they are essential to improve surveillance, for monitoring transmission, disease burden and outbreaks, or to evaluate control measures. Considering diagnostics directly inform several targets set in the 2021–2030 roadmap, WHO established in 2019 a Diagnostic Technical Advisory Group for NTDs (DTAG-NTD) to address priority areas for NTD diagnostics, identify gaps in access, and advise on developments required for the diagnostic process to properly inform decisions on NTD treatment [5, 6].

WHO's 2021–2030 roadmap assessment of the current scenario for VL diagnosis is that (i) it lacks adequate diagnostics for surveillance, (ii) major changes are required on diagnostic tests available for screening and diagnosis confirmation, (iii) sensitivity of rapid tests is insufficient in certain regions, (iv) a viable test of cure for VL and PKDL is absent, and that (v) PCR is restrict to reference laboratories [4]. The specific priorities set for VL in the 2021–2030 roadmap were to develop: (i) more sensitive rapid diagnostic tests for use in East Africa, (ii) less invasive, highly specific tests to measure parasite level, and (iii) a minimally invasive test of cure for PKDL and VL. The following critical actions were defined: (i) enable early detection to ensure timely treatment, and (ii) develop more effective and user-friendly diagnostics, especially for East Africa [4]. The scientific literature adds to that the need for more reliable VL tests to detect acute phase, relapse, and *Leishmania*-HIV coinfection [7].

Target Product Profiles (TPPs) for point of care *in vitro* detection of active VL and for *in vitro* laboratory-based test of cure for VL post-treatment have recently undergone public consultation. Draft versions of the TPPs highlight the need of diagnostic tests with the following ideal features: detection of analytes specific to *L. donovani* or *L. infantum*, with more than 95% clinical sensitivity and 99% clinical specificity; qualitative result for detection of active VL and quantitative result to confirm VL cure; execution under zero-infrastructure conditions, at low cost, using peripheral whole blood from finger stick, urine or saliva as samples, and enabling test result within 30 minutes [8].

The roadmap assessment for CL is that (i) current diagnosis based on parasitological tests and/or clinical features is not sufficiently sensitive in several endemic areas and laboratory diagnosis is not always available, and (ii) PCR is only available in reference laboratories. Improving the affordability and sensitivity of rapid diagnostic tests available at the health center and community levels, including detection at species level, is one of the key actions identified for CL. This is especially important in foci where multiple *Leishmania* species coexist [4]. It should be emphasized that identification of the responsible *Leishmania* species can be crucial to prognosis, disease control and therapeutic interventions [9, 10].

A 2019 TPP for a point of care test for dermal leishmaniasis considers the following specific optimal features: species-specific detection of any form of CL or PKDL with more than 95% specificity, the use of *Leishmania* antigens as targets, direct testing from lesion swab, and less than 20 minutes to result, with visual reading. Minimal features are genus-specific detection of active localized CL with more than 90% specificity, using aspirates/biopsies/skin scrapings as samples, and obtaining results in less than 1h upon visual reading or using a simple reading device [11].

Although several reviews describe recent advances in leishmaniasis diagnosis published in the scientific literature [9, 12–16], we could not find recent patent landscape analyses on this subject. Patent landscapes give a snapshot of innovation in a technological field of choice at a given time, providing insight into the patenting dynamics, key players, R&D location, markets of protection, technological trends, emerging technologies, among others. Such analyses are a useful tool to support decision making and R&D investment.

The current study aims to contribute to the understanding of leishmaniasis diagnosis by providing an overview of the patent filings in this field and analyzing whether the developments revealed in these documents are consistent with the needs described by the scientific community, in special the main gaps detected by WHO's roadmaps. We focus on patent documents disclosing experimental evidence of diagnostic use submitted within the past decade, rather than patent documents that claimed a diagnostic test for leishmaniasis but did not necessarily support such claims experimentally. This strategy allows for a comprehensive and more informative analysis of the diagnostic method disclosed. A limitation of this approach is that evidence for leishmaniasis diagnosis may be obtained after the patent application, in which cases relevant inventions will be ignored. Nonetheless, we believe that the overestimation resulting from including all documents, regardless of experimental evidence, is much more detrimental than the possible underestimation that could result from the use of an experimental evidence filter. This patent landscape can be used to inform policy discussions, guide direct investment and strategic research planning.

## Materials and methods

This patent landscape follows OECD's Patent Statistics Manual guidelines [17] and the checklist for Reporting Items for Patent Landscapes (RIPL) [18].

### Dataset compilation S1 Data

**Search scope and strategy.**   Searches were carried out in September 2022 using the proprietary database Orbit Intelligence (Questel, Paris, France). This database covers patent documents published by more than 100 patent authorities worldwide, encompassing more than 110 million patent publications at the time of writing. Our search strategy targeted patents filed after 2010 disclosing a method for diagnosing leishmaniasis. Therefore, we only included patent families for which the first patent application was filed after 01/01/2010, i.e., documents with earliest priority after this date. More specifically, we first searched for documents containing the words leishmania+ and (+diagnos+ or detec+) in their title, abstract, or claims. We also used the Cooperative Patent Classification (CPC) and International Patent Classification (IPC) codes to include documents containing the word leishmania+ in their title, abstract, or claims and classified as (i) "immunoassay or other binding assay" (under CPC or IPC G01N33/569), (ii) "a measuring/testing process involving enzymes, nucleic acids or microorganisms" (under CPC or IPC C12Q1+) or as (iii) a "test involving the analysis of chemical or physical properties" (under CPC or IPC G01N+); or containing the word leishmania+ in the claims and classified as "peptides derived from protozoa" (under CPC or IPC C07K14/44); or containing the word diagnos+ in their title, abstract, or claims and classified as "*Leishmania* antigens" (under CPC or IPC A61K39/008). See S1 Text for the specific search strings used.

**Patent grouping into families.**   Documents retrieved by our searches were automatically grouped by Orbit Intelligence into FamPat patent families. Orbit rules for FamPat construction are designed to group together all patent publications related to a single invention, such as different stages of a patent application in a particular country or related applications filed in different countries.

**Patent selection criteria.** Inventions outside the scope of our search (i.e., unrelated to leishmaniasis diagnosis) and inventions within our search scope but which did not show experimental evidence of leishmaniasis diagnosis were manually excluded. At this stage some patents classified by Orbit into different patent families were grouped as a single family, when they protected the same invention and had a priority document linking the family. Manually grouped families are listed in S2 Text. In addition, two patent documents from Brazil were excluded from our dataset because they were continuation of documents with priority dates before 01/01/2010.

**Data extraction.** The following data was automatically extracted from the FamPat families: current standardized assignee, assignee country, assignee address, inventor name, inventor address, earliest priority date, family legal status (pending, granted, revoked, expired, lapsed), family legal state (alive, dead), countries/authorities. Patents families were automatically ungrouped to obtain individual patent application numbers and the legal state of individual filings (FullPat records).

## Manual classification of data

**Extracted data standardization.** Data was manually cleaned to harmonize assignee name and remove from this field the funding agency's, university board of regents', or technology transfer office's name.

**Assignee classification.** Assignees were classified as "Academy" (universities, research institutes, government, and other not-for-profit entities), "Corporate" (companies), or "Individuals" (where an individual was indicated as assignee without affiliation to any organization).

**R&D collaborations.** The field "current standardized assignee" was used to determine whether the invention was developed in collaboration. When the assignee was an independent inventor, this classification was not considered applicable. When developed in collaboration, inventions were further classified to indicate whether they consisted in academy-only, corporate-only, or academy-corporate partnerships.

**Classification of diagnostic methods.** Patent documents were thoroughly analyzed and classified based on the experimental evidence disclosed. It was determined whether the test consisted of a molecular or immunological method, the precise method used (e.g., PCR, RT-PCR, ELISA etc.), the antigen or target gene employed, the clinical presentation under investigation (VL, CL etc.), *Leishmania* species tested, test sample, and whether it demonstrated detection of asymptomatic infection or detection of leishmaniasis in individuals coinfected with HIV. For molecular methods, we also specified if species typing and parasite quantification were demonstrated. In the case of immunological methods, we further indicate whether the antigen is identical to its natural form (or parts of it) or whether it has been modified. The presence of accuracy results and level of evidence were analyzed based on [19], that being: 1. An independent, masked comparison with reference standard among an appropriate population of consecutive patients; 2. An independent, masked comparison with reference standard among nonconsecutive patients or confined to a narrow population of study patients; 3. An independent, masked comparison with an appropriate population of patients, but reference standard not applied to all study patients; 4. Reference standard not applied independently or masked; 5. Expert opinion with no explicit critical appraisal, based on physiology, bench research.

## Patent data analyses

**Patent timeline.** *Patent counts*. To obtain an overall picture of inventive activity, patent family counts were plotted by earliest priority year. Earliest priority year was chosen as the

closest date to the invention and best indicator of inventive performance, following OECD's recommendations [17].

*Family size.* The number of different authorities in which patents from each patent family were filed was assessed using Orbit's field "Countries/authorities count". This field was corrected for the patent families that were manually grouped. This is considered an indication of investment in the protection of each invention, as additional fees are required for each filing in a different authority.

**Identification of R&D country.** The assignee country was used to infer where R&D activity took place. Inventors' address was used when no assignee address was available or when further clarification was needed. When the above-mentioned information was not available, we Google searched assignee name to ascertain its location. For a small number of patents R&D location could not be identified by any of the previous strategies and priority country was used as R&D location.

**Markets of protection.** For this analysis we considered the countries where patents are still alive, either granted or pending. To obtain this information, patent families were automatically ungrouped (Orbit's Fullpat records). The individual patents were filtered by patent status and the country codes of alive patent documents were extracted.

**Patent status.** A family is considered alive if it has at least one live member, either granted or pending. Thus, to count it as dead, all members must be dead.

**Assignee type and collaboration.** The following was assessed: (i) the number of patent families per assignee type ("academy", "corporations" or "individuals"); (ii) how many of these families are jointly owned by two or more assignees; and (iii) the precise type of collaboration (academy-only, corporate-only, or academy-corporate). When more than one individual assigned a patent family it was not considered a collaboration, as in many of such cases the patent family will be later reassigned to an academic institution or corporation where the invention was developed.

**Top assignees.** A list of all patent assignees (n = 45) was gathered and the number of times each name is indicated as patent family assignee was assessed (totaling 101 occurrences). Eight assignees were individuals and were not considered in the present analysis.

## Results

Our search strategy resulted in the retrieval of 423 patent families. Each patent family contains one or more individual patent applications related to a single invention. Multiple filings under the same family often correspond to filings for the same invention in different countries. Of these 423 patent families, only 94 showed experimental evidence of leishmaniasis diagnosis (amounting to a total of 136 individual patent applications). These are the patent families that disclose a diagnostic method for leishmaniasis with experimental support first filed after 01/01/2010. Mere identification of immunogenic proteins without confirmation of diagnostic potential, by western blot for instance, was considered insufficient to meet the inclusion criteria. All our analyses are based on this specific set of patent families.

### Patenting dynamics

Patenting activity for leishmaniasis diagnosis is low. In the most active year only 11 patents were filed, whereas lowest activity occurred in 2011 (4 patents filed). Given that most patent authorities publish patent applications up to 18 months after filing, the drop in filings observed in the last couple of years is expected (Fig 1A). An average of 8 filings per year is detected when the last two years are removed. In terms of patent family size, the vast majority (78%) of patent families consist of single patent applications, while 13% of patent applications were filed

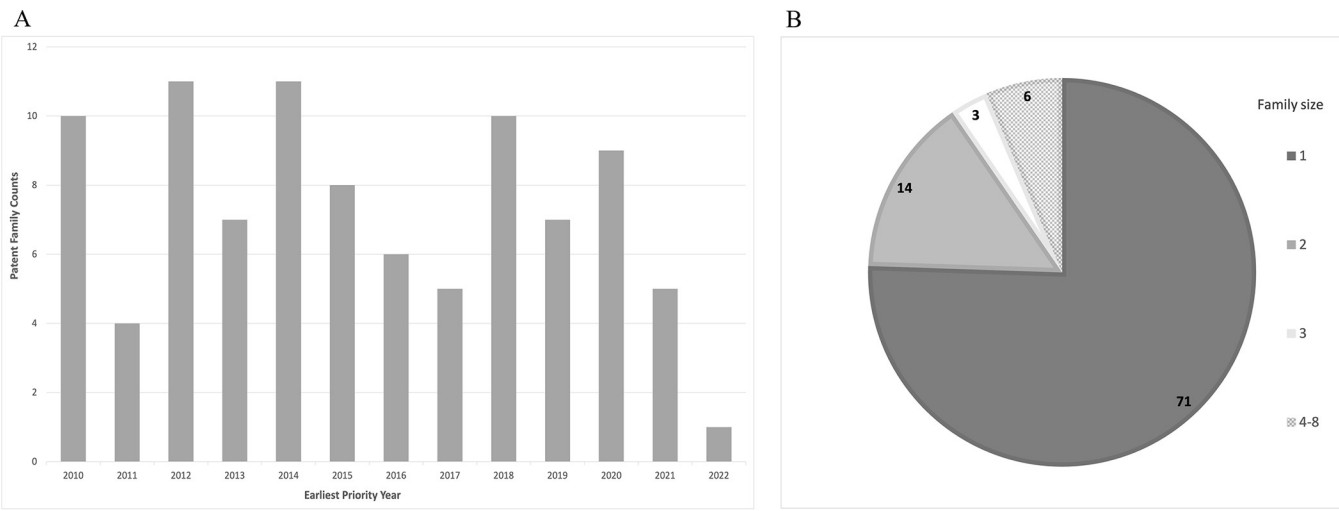

**Fig 1. Patent family counts and size.** (A) Patent family counts. Each patent family is counted once, in the year the first patent in the family was filed i.e., the earliest priority year. (B) Patent family size. Patent families are accounted for by size (the number of different authorities in which each patent family was filed).

in two patent offices. Only 9% of patent families were filed in three or more patent authorities (Fig 1B).

Most R&D activity (59%) took place in Brazil, followed by China (17%), the USA (7%) and India (6%). The remaining countries have smaller contributions (2% or less each) (Fig 2A). Regarding the actual markets of protection, most live patents are in force in Brazil (54%), China (13%), India (11%) and the US (10%) (Fig 2B).

The previous patent family counts include applications that are alive, either pending examination or granted, but also applications that are already dead, i.e., abandoned by the assignee, expired, or revoked. Patent legal status was analyzed to ascertain how many of the 94 patent families currently protect inventions or still have the potential to protect them. Our results showed that 70% of the patent families are alive (i.e., they have at least one live member, either granted or pending) (Fig 3). From these live families, most (62%) are pending applications, whereas 38% contain at least one patent in force (i.e., granted).

Assignees were further classified according to the institutions supporting the inventive activity. Academic institutions appear as assignees in 80% of the patent families, corporations in 12% and individuals in 8% (Fig 4). Only 16% of our patent families had more than one academic institution or corporation as assignee. Of these, 86% are co-assigned by the academic sector, 7% by corporations, and 7% by the academy and corporations.

Top applicants are mostly Brazilian academic institutions. Universidade Federal de Minas Gerais (UFMG) is by far the institution with the highest number of patents, assigning 33% of the patent families, followed by Universidade Federal de Uberlândia (UFU), Fundação Oswaldo Cruz (Fiocruz) and Universidade Federal do Paraná (UFPR) (assigning 6%, 5% and 5%, respectively). The institutions represented in Fig 5 are the only ones that own 2 or more patent families from the 45 institutions assigning patents in our dataset.

## Assessment of experimental evidence

A thorough analysis of experimental evidence contained in the 94 patent families of our dataset indicated that 61 of them disclosed immunological methods for leishmaniasis diagnosis whereas 32 disclosed molecular methods. One method that did not fit either category was classified as "other". A total of 63 immunological and 33 molecular methods were revealed in

A

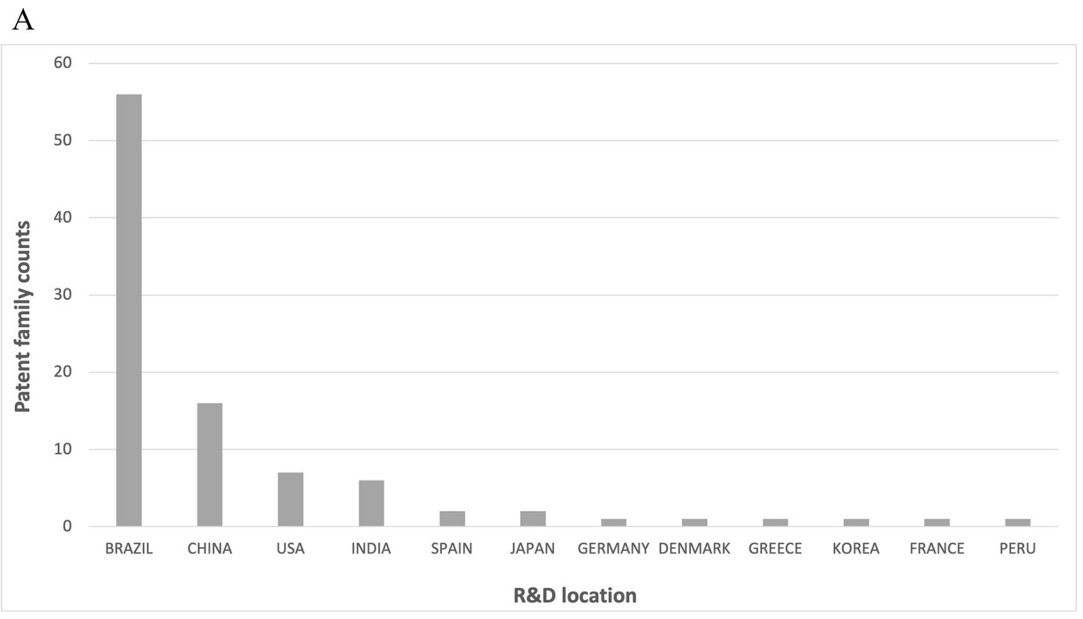

B

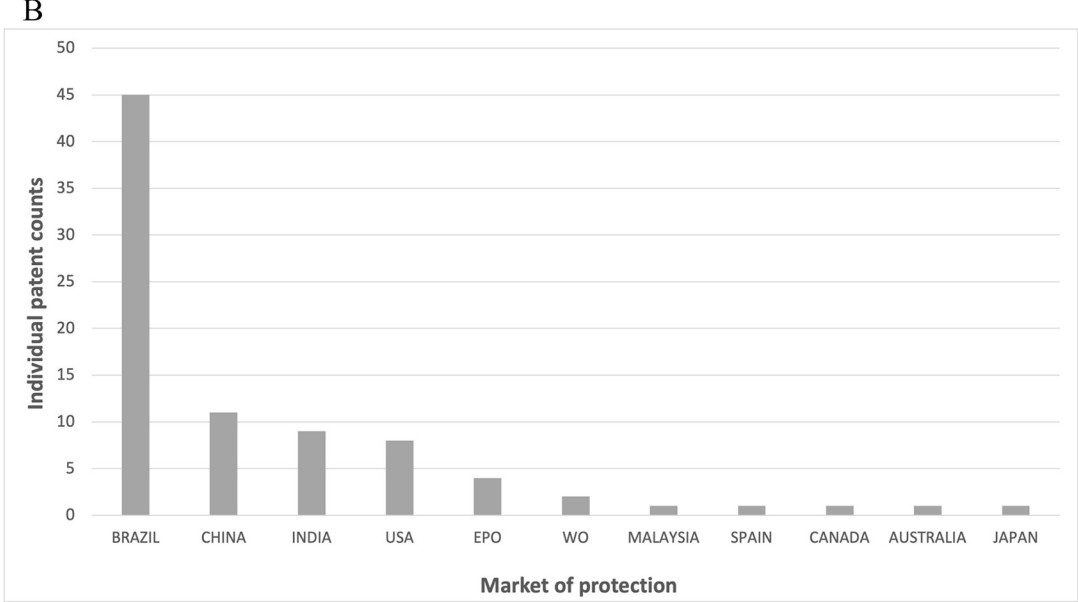

**Fig 2. R&D location and markets of protection.** (A) Patent family counts are assessed by R&D location. Patent family assignee country is used as an indication of R&D location. When assignee country was unknown, inventor's address was used instead. (B) The country codes of live individual patent filings (Fullpat) are accounted for. Patents filed in Europe (EPO) or via PCT (WO) that can still enter the national phase are included.

these patent families, as some of them presented experimental evidence of more than one mode of implementing the invention. Most patents reveal a diagnostic method based on ELISA assays (50) or Polymerase Chain Reaction (PCR) (20). The specific methods disclosed are summarized in Table 1.

**Molecular methods.** A more detailed analysis of patent families disclosing molecular methods is summarized in Table 2. Most of them target the kDNA minicircle (11) or 18S ribosomal RNA/DNA (4). Considering the form of the disease and species tested, most target VL (23) and TL (18, including CL data). The most tested species are *L. donovani* (19), *L. infantum*

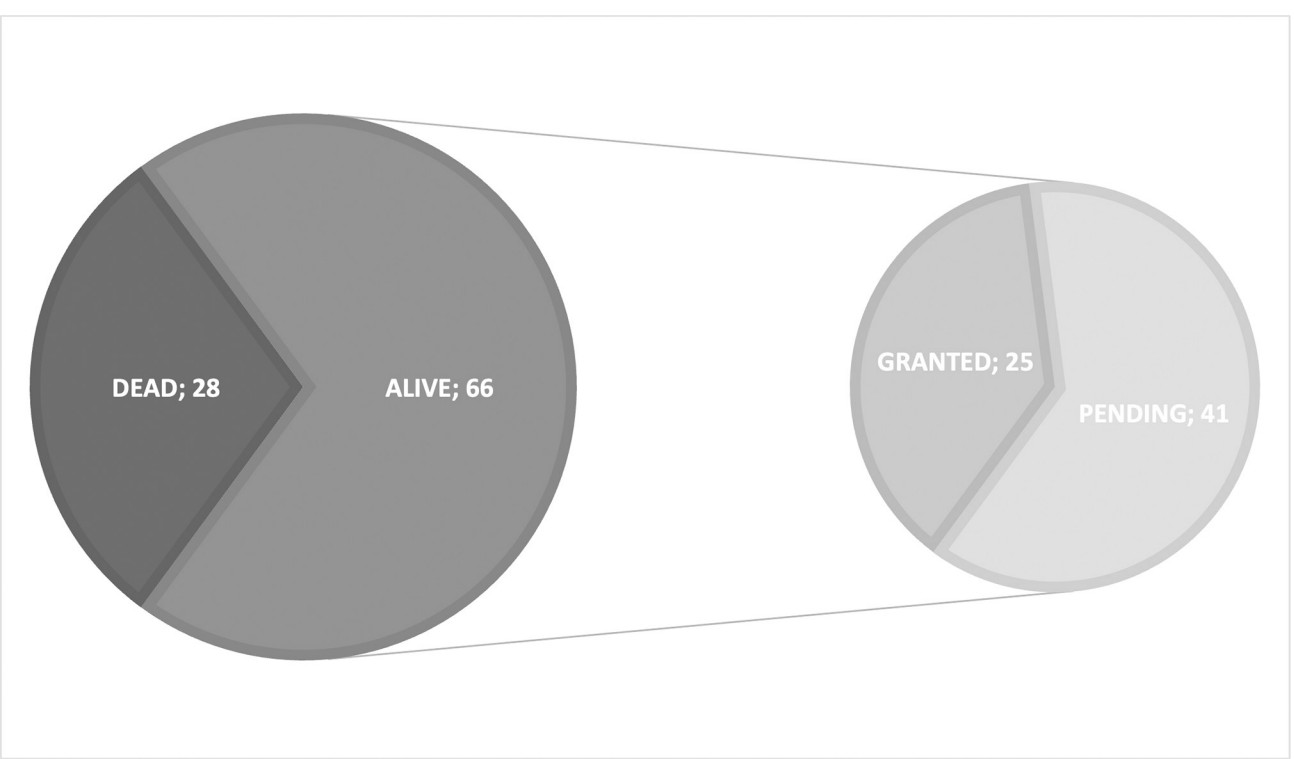

**Fig 3. Patent family status.** Families are considered alive if they have at least one member still in force. When the live family contains at least one granted patent, the whole family is classified as granted. Otherwise the family is regarded as pending, indicating that applications belonging to this family are still under review by the respective national patent office(s).

(16) or *L. major* (15). *Leishmania* quantification was demonstrated in 14 of these 32 families and another 14 evidence typing, though in one case at complex level only (not at species level). Seven families included experimental evidence demonstrating both quantification and species typing. Only one of the tests was assessed for detection of asymptomatic individuals and none for HIV/*Leishmania* co-infection. Clinical validation is included in 15 of the patent families. From these, 12 are validated using human samples only, one used dog samples only, one used both human and dog samples, and one used human, dog, and insect vector. Only three of them presented accuracy results, one classified as level of evidence 4 and the other two as level 2 (based on Turlik, 2009 [19]).

**Immunological methods.** A more detailed analysis of patent families disclosing immunological methods is summarized in Table 3. Overall, there is a predominance of tests based on the detection of antibodies. Most tests are based on K39 or on a K39-homologue (12), either alone or in combination with other antigenic regions. Thirty-four of the patent families do not employ the whole protein as target antigens, but selected peptides and epitopes (alone or combined), 6 of which are chimeric proteins. Considering the form of the disease and species tested, most target VL (53) and TL (22, including CL data) and most tests are carried out with *L. infantum* (39) or *L. braziliensis* (19). From the 61 patent families, only 17 confirmed detection of asymptomatic individuals and 3 detected leishmaniasis in cases of HIV/*Leishmania* co-infection. Clinical validation is included in 56 of the patent families. From these, 19 are validated using human samples only, 21 use dog samples only, and 16 use both human and dog samples. Accuracy results are included in 44 of them, 43 of which are classified as level of evidence 4 and one as level 2.

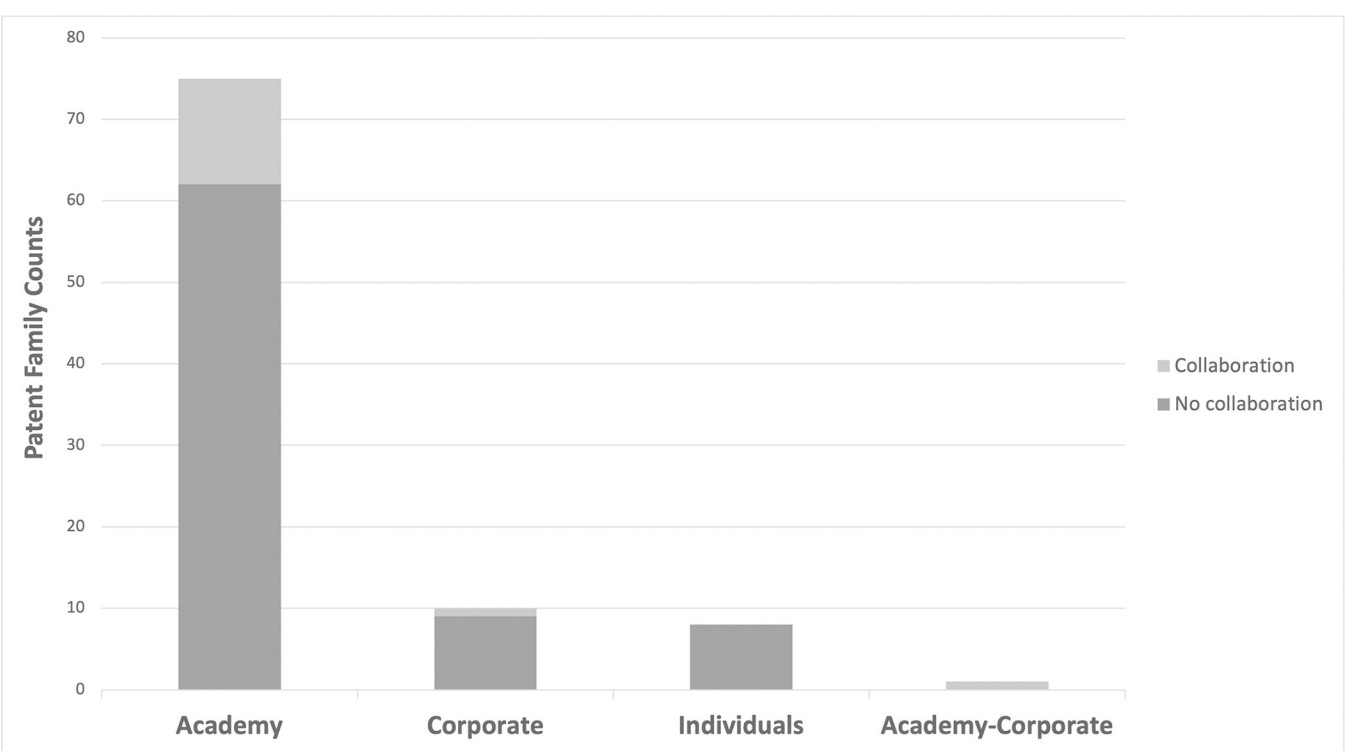

**Fig 4. Classification of assignee counts by assignee type.** Assignees were classified as "Academy" (universities, research institutes, and other not-for-profit entities), "Corporate" (companies), and "Inventor" (individual without affiliation to any organization). The number of patent families having each of these assignee types is shown. The number of patent families with a single assignee are represented in blue, whereas those with two or more assignees (indicating a collaboration) are depicted in orange. Patents assigned by individuals were considered as being owned by a single assignee.

**Other methods.**    One method did not fit into the categories of molecular or immunological method and was classified as "others" (Table 4).

## Discussion

Our results suggest very slight worldwide interest in patenting leishmaniasis diagnostic methods, with few patents filed per year (mostly protecting the invention in a single country). Although diagnosis is an essential component of any NTD control program, from disease confirmation to mapping, screening, surveillance, monitoring and evaluation, diagnostics are overall a neglected area in healthcare, receiving little attention and funding [20]. As patenting is a relatively expensive process, especially when protection is sought in multiple jurisdictions, the low interest in leishmaniasis diagnosis is not surprising.

Most applications come from institutions in Brazil, followed by China, India, and the USA. These are also major markets of protection. This finding is consisting with the outstanding contribution of Brazil, the USA and India to leishmaniasis research [21–25], the fact that these countries are endemic for leishmaniasis [26] and that China and the USA are the leading countries in overall patent filings and filings by residents [27]. The expressive contribution of Brazil and India is also in line with the impact of the disease in these countries. According to data from the Global Health Observatory, India and Brazil are among the top five countries in reported VL cases from 2005–2020. China appears in the 12th position. Brazil is also among the top three countries in reported CL cases from 2005–2020 [28]. Most surprising is the lack of patents from the United Kingdom, Iran, Colombia, Venezuela and Spain, countries that

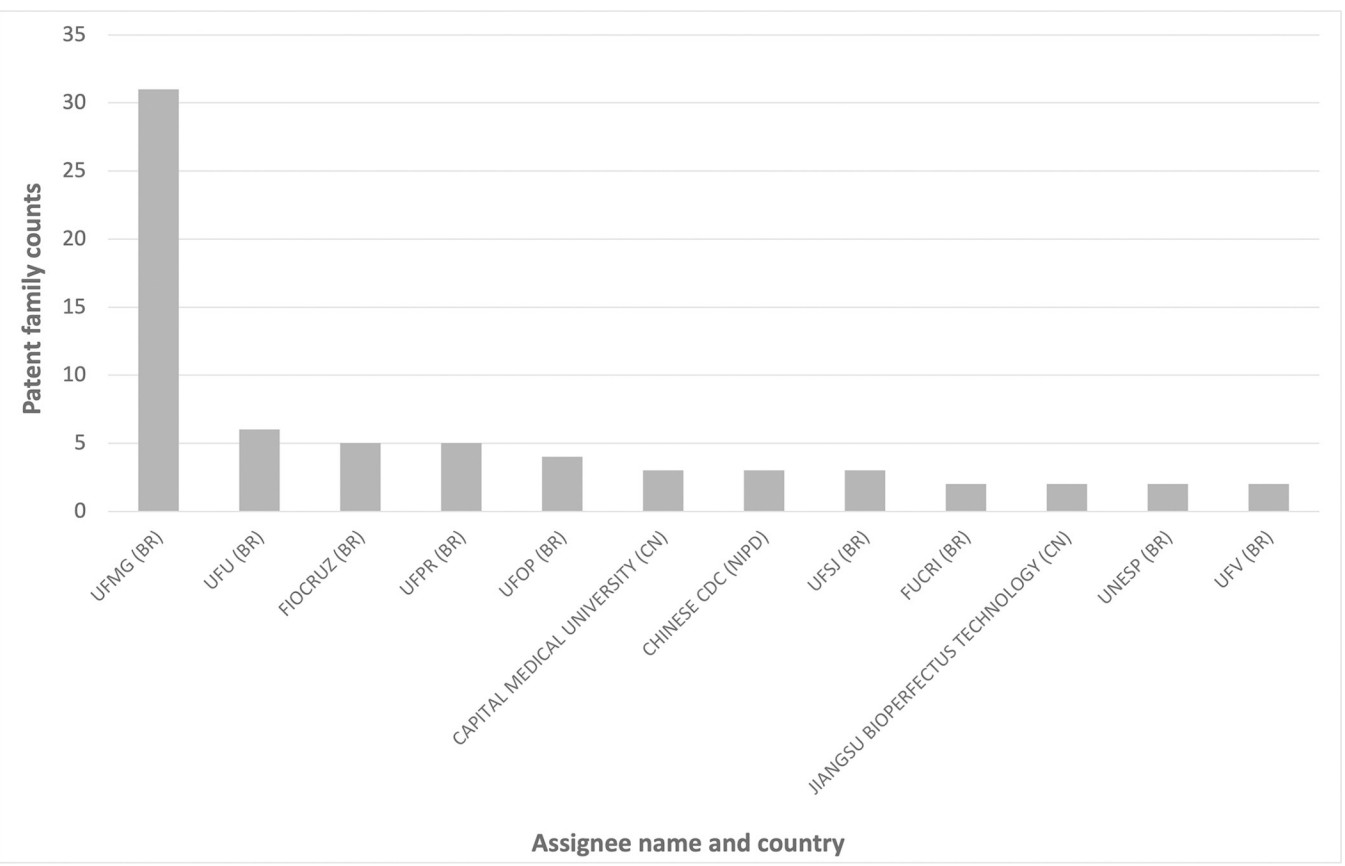

**Fig 5. Top applicants.** The number of times each assignee name is indicated as patent family assignee is represented. Legend: UFMG- Universidade Federal de Minas Gerais, UFU—Universidade Federal de Uberlândia, Fiocruz—Fundação Oswaldo Cruz, UFPR—Universidade Federal do Paraná, UFOP—Universidade Federal de Ouro Preto, Chinese CDC—Chinese Center For Diseases Control & Prevention, UFSJ—Universidade Federal de São João Del Rei, FUCRI—Fundação Educacional de Criciuma, UNESP—Universidade Estadual Paulista Júlio de Mesquita Filho, UFV—Universidade Federal de Viçosa.

significantly contribute to scientific research in the field, some of which are also affected by leishmaniasis [21, 29].

Patent grant can be used as a quality indicator of innovation activities [30]. However, considering almost half of the 94 families are still under examination by patent offices, it is not known whether the invention disclosed in these applications are in fact new and inventive compared to the prior art [30].

According to our analyses, the R&D behind patenting is almost entirely carried out by universities and research institutions, with little contribution from companies. Such low private sector interest is expected, given the low financial return from interventions primarily targeting low-income populations. Universidade Federal de Minas Gerais (UFMG) is by far the leading assignee with 33% of patent families. Notably, the top five assignees, owing 54% of patent families, are all Brazilian universities/research institutions. Out of 45 assignees, only 11 hold two or more patent families. From these, nine are Brazilian and two are Chinese universities/research institutions.

Our patent data analyses indicate that the strong international collaborative research observed when analyzing scientific publications on leishmaniasis worldwide [23, 25] does not lead to co-ownership of patent applications for leishmaniasis diagnosis. In fact, only 16% of our patent families contained some type of joint research (international or national), the vast

**Table 1. Method disclosed in the patent families by type.**

| Diagnostic test type | | Patent family counts |
|---|---|:---:|
| **Molecular** | | **33** |
| PCR based | Real-time PCR (RT-PCR) | 6 |
| | PCR simplex | 4 |
| | Multiplex PCR | 3 |
| | PCR followed by sequencing | 3 |
| | Multiplex RT-PCR | 2 |
| | Real time PCR followed by High Resolution Melting (HRM) analysis | 1 |
| | PCR followed by hybridization detection | 1 |
| LAMP-based | LAMP | 4 |
| | Multiplex RT-LAMP | 2 |
| RPA-based | Recombinase polymerase amplification (RPA) | 2 |
| | RPA followed by electrochemical detection | 1 |
| Molecular detection without DNA amplification | | 2 |
| Recombinase-aided amplification (RAA) | | 1 |
| Strand displacement amplification (SDA) | | 1 |
| **Immunological** | | **63** |
| ELISA | | 50 |
| Immunochromatographic test (ICT) | | 4 |
| Flow cytometry | | 3 |
| Biosensor | | 2 |
| Latex Agglutination Assay (LA) | | 1 |
| Crosslink immunoprecipitation (CLIP) | | 1 |
| Whole blood assay | | 1 |
| Skin test | | 1 |
| Other methods | | 1 |
| Gas chromatography/mass spectrometry | | 1 |

The number of patent families disclosing each type of diagnostic method for leishmaniasis is represented.

majority of which occurred between two academic institutions from the same country. It seems that either collaboration is low for leishmaniasis in general, or it is more focused on basic research and other research topics rather than diagnostics, such as drug or vaccine development.

More detailed analysis of the methods disclosed in the patent documents showed that kDNA is the most used target region in the molecular method patent families. kDNA is often targeted for its abundance, specificity, and repetitiveness. A drawback of using this gene to quantify parasites is the uncertainty of whether kDNA copy number differs between *Leishmania* species, strains, and growth stages. While seven of the patent families disclosing a kDNA-based test give experimental evidence of parasite quantification, only three mention the *Leishmania* species used and only one of these uses more than two species. Therefore, it is not possible to ascertain that all seven kDNA based tests will quantify parasites regardless of *Leishmania* species, strain, and growth stage.

Our results indicate that K39 or K39-homologues are the most popular target genes in the patent families disclosing immunological tests (either alone or in combination with other antigenic regions). K39 is an antigen used in commercial tests. The efficiency of current rapid diagnostic antibody detection tests based on K39 varies by region. For instance, while 98% of patients with primary VL in South Asia can be diagnosed with such tests, this number drops to

**Table 2. Patent families disclosing molecular methods.**

| Patent number | Assay type | Target | Test with clinical sample | Accuracy results/ Level of evidence* | Details | *Leishmania* species tested |
|---|---|---|---|---|---|---|
| BR102017017125 | PCR | Lc36 (TGMACK) | No | No (n/a) | Typing | VL (*L. infantum*) |
| IN201002940 | PCR | 18S ribosomal DNA | Yes (H) | Yes (2) | - | VL, PKDL (*L. donovani*) |
| CN101921862 | PCR | ITS2 | No | No (n/a) | Sandfly-only | n/a |
| BR102019026180 | PCR/RT-PCR | DNA polymerase catalytic subunit A | No | No (n/a) | Hamster sample; Typing | VL, TL (*L. braziliensis, L. amazonensis, L. infantum*) |
| CN106319060 | Multiplex PCR | n/a | No | No (n/a) | - | TL (*L. major*) |
| BR102018003443 | Multiplex PCR | microsatellite region and orthologue gene | No | No (n/a) | Typing | VL (*L. donovani, L. infantum*); TL (*L. guyanensis* complex, *L. amazonensis, L. braziliensis, L. mexicana*) |
| BR102012026282 | Multiplex PCR | SL mini-exon RNA | Yes (H, C) | No (n/a) | Typing C | VL (*L. donovani* complex); TL (*L. braziliensis, L. mexicana*) |
| CN110734995 | RT-PCR | kDNA minicircle† | Yes (H) | No (n/a) | Quant | Species not mentioned |
| CN111549160 | RT-PCR | kDNA minicircle† | No | No (n/a) | Quant; Typing | VL (species not mentioned) |
| US10883147 | RT-PCR | kDNA minicircle, Mag I, DNA pol 1, DNA pol2, HSP70, Cyt B, mini-exon | Yes (H) | No (n/a) | Quant; Typing; asympt | VL (*L. donovani, L. infantum*); TL (*L. tropica, L. major, L. mexicana, L. amazonensis, L. braziliensis*) |
| CN112795676 | RT-PCR | HGPRT | Yes (H) | No (n/a) | Quant; Typing | VL (*L. donovani, L. infantum*) |
| CN112795677 | RT-PCR | SPDSYN | Yes (H) | No (n/a) | Quant; Typing | TL (*L. major, L. tropica, L. donovani, L. infantum*) |
| CN113897448 | Multiplex RT-PCR | ITS-1, ITS-2, P0, RACK | Yes (H) | No (n/a) | Artificially contaminated samples; Quant; Typing | VL (*L. donovani, L. infantum*); CL (*L. tropica, L. major*) |
| CN114574607 | Multiplex RT-PCR | kDNA minicircle | Yes (H) | No (n/a) | Quant | VL (species not mentioned) |
| BR102016018960 | RT-PCR-HRM | HSP70 | Yes (H, C, I) | No (n/a) | Quant; Typing | VL (*L. donovani, L. infantum*); CL (*L. tropica, L. major, L. amazonesis, L. mexicana, L. lainsoni, L. braziliensis, L. guyanensis, L. naiffi, L. shawi*); PKDL (*L. donovani*) |
| CN103409502 | PCR-hybridization | 28s rRNA DNA spacer sequence | Yes (H) | No (n/a) | - | TL (species not mentioned) |
| CN111621583 | PCR-seq | 6PGD, LACK, ASAT, G6PD | Yes (H) | No (n/a) | Typing; lesion | VL (*L. donovani, L. infantum*); CL (*L. major*) |
| WO201471946 | PCR-seq | 18S ribosomal DNA | No | No (n/a) | - | n/a |
| CN106047993 | PCR-seq | putative Lanosterol synthetase† | Yes (H) | No (n/a) | - | VL (*L. donovani*) |
| CN111876512 | LAMP | kDNA minicircle | No | No (n/a) | Quant; Typing | VL (*L. donovani, L. infantum*) |
| IN327506 | LAMP | kDNA minicircle | Yes (H) | Yes (4) | - | VL and PKDL (*L.donovani*); TL (*L. tropica, L. major*) |
| BR102019005228 | LAMP | kDNA minicircle† | No | No (n/a) | Hamster samples; Quant | TL (*L. amazonensis*) |
| WO2022109690 | LAMP | HSP70 | Yes (H) | Yes (2) | - | VL (*L. donovani, L. infantum*); TL (*L. braziliensis, L. amazonensis, L. guyanensis, L. lindenberg, L. panamensis, L. hertigi, L. naiffi, L. shawi, L. major, L. mexicana*). |
| US10072309/ US20200048722 | Multiplex RT-LAMP | 18S ribosomal RNA† | No | No (n/a) | Quant | Species not mentioned |
| WO2020050852 | Multiplex RT-LAMP | 18S ribosomal RNA† | No | No (n/a) | Quant | Species not mentioned |
| US2016130669/ WO201506755 | RPA | kDNA minicircle | No | No (n/a) | Typing | VL (*L. infantum*); TL (*L. major, L. braziliensis*) |

*(Continued)*

**Table 2.** (Continued)

| Patent number | Assay type | Target | Test with clinical sample | Accuracy results/ Level of evidence* | Details | *Leishmania* species tested |
|---|---|---|---|---|---|---|
| CN108588252 | RPA | kDNA minicircle | No | No (n/a) | Typing | VL (*L. infantum*, *L. donovani*); TL (*L. braziliensis*, *L. major*, *L. mexicana*) |
| EP3167075 | RPA-electrochemical detection | kDNA minicircle† | Yes (C) | No (n/a) | Quant | VL (*L. infantum*) |
| WO2012124681 | SDA | unidentified tandem repeat | No | No (n/a) | - | VL (*L. donovani*) |
| CN113755620 | RAA | ITS-1 | No | No (n/a) | Quant | VL (*L. donovani*, *L. infantum*) |
| EP2631300 | Detection without DNA amplification | kDNA minicircle | No | No (n/a) | - | VL (*L. infantum*, *L. donovani*); TL (*L. tropica*, *L. major*) |
| US8975390 | Detection without DNA amplification | n/a | No | No (n/a) | - | VL (*L. donovani*); TL (*L. major*) |

**Codes:** †Identified by Blast; Quant—quantitative assay; Typing—typing demonstrated at species level; Typing C—Typing demonstrated at complex level; asympt—detects asymptomatic infection; ac—artificially contaminated samples; n/a—information not available. Ind inventor—independent inventor. **Gene codes:** ASAT—Aspartate aminotransferase; Cyt B—Cytochrome B; G6PD—Glucose-6-phosphate 1-dehydrogenase; HGPRT—Hypoxanthine phosphoribosyl transferase; ITS—Internally Transcribed Region; LACK—*Leishmania*-activated C-kinase antigen; MAG1- MSP Associated Gene I; RACK—receptors for activated C kinase; SL mini-exon—Spliced leader mini-exon RNA; SPDSYN—Spermidine synthase; 6PGD—6-phosphogluconate dehydrogenase. *Level of evidence based on Turlik, 2009 [19].

85–90% in East Africa [31]. Many of the patent families disclosing a K39 (or homologue)-based test claim that their goal is to provide an improved test for human or canine VL. However, only five patent families included comparisons with K39 and demonstrated that higher sensitivity is obtained.

Regarding gaps identified in WHO's 2021–2030 roadmap [4], none of the patent families in the current landscape include enough experimental evidence to validate a rapid test. Moreover, only half of the molecular patent documents retrieved presented evidence of species typing, less than half presented some evidence of parasite quantification and only two patent families show the validity of the test on patients from East Africa or using DNA samples from East Africa isolates. Regarding the request of a test of cure for VL and PKDL, while some documents retrieved indicated that the disclosed diagnostic method could be used to differentiate healthy from treated individuals, only one family discloses a possible test of cure for VL supported by experimental evidence, and none presented validation for PKDL. In fact, experimental demonstrations with PKDL patient samples are only available in four patent families. This could possibly be explained by the difficulty in obtaining individual samples prior and after treatment, the fact that patents are filed early in the R&D process to guarantee priority date, and/or that PKDL is most common in East Africa and South-East Asia while most R&D work behind these patents took place in Brazil.

HIV–*Leishmania* coinfection and identification of asymptomatic dogs are important topics to be addressed if we are to achieve the goal of eliminating VL as a public health problem. HIV infection poses a major threat to leishmaniasis control and increases the risk of developing VL by more than 100-fold [32]. Despite the clear need for a rapid test capable of detecting *Leishmania*-HIV coinfection, only three patent families demonstrate test efficacy in such case. Identification of asymptomatic dogs is included in 18% of patent families, all by immunological methods. Most use canine samples to validate the test.

Regarding test format, health professionals should be able to perform such tests in difficult field conditions without the need of specific scientific expertise. Affordability is an important

**Table 3. Patent families disclosing immunological methods.**

| Patent # | Assay type | Target | Antigen details | Test with clinical sample | Accuracy results/ Level of evidence* | Test details | *Leishmania* species tested |
|---|---|---|---|---|---|---|---|
| IN2011DE03380 | CLIP | Unidentified | Peptide purified from patient urine | Yes (H) | No (n/a) | Antigen detection in urine | VL (*L. donovani*) |
| WO2016113749 | ELISA | LAg | n/a | Yes (H) | Yes (4) | Antibody detection in urine | VL (*L. donovani*) |
| WO2022150899 | ICT | DTL4 (A2, K39) | Chimera | Yes (H, C) | Yes (4) | HIV+ | VL (*L. infantum*) |
| IN2011DE03379 | Whole Blood Assay | SLA | Soluble antigens | Yes (H) | Yes (4) | asympt | VL (*L. donovani*) |
| BR201000664 | ELISA | *Leishmania* whole extract | *Leishmania* extract | Yes (C) | Yes (2) | - | VL (*L. infantum*) |
| BR102012030066 | ELISA | Perodoxin | Recombinant protein | Yes (C) | Yes (4) | - | VL (*L. infantum*) |
| WO201529002 | ELISA | A2 | Recombinant protein with codon optimization | Yes (C) | No (n/a) | - | VL |
| BR201013447 | ELISA | A2 and K39 | Epitope combination | Yes (H, C) | Yes (4) | asympt | VL |
| BR201005033 | ELISA | A2, NH and LACK | Epitope combination | Yes (H, C) | Yes (4) | asympt | VL |
| BR102017022744 | ELISA | K39 homologue | Recombinant peptide | Yes (H) | No (n/a) | - | VL (*L. infantum*); CL (*L. braziliensis*) |
| BR102020009366 | ELISA | K39 homologue | Synthetic peptide | Yes (H, C) | Yes (4) | - | VL (*L. infantum*); CL (*L. braziliensis*) |
| BR102020007615 | ELISA | K39 homologue | Synthetic peptide | Yes (C) | Yes (4) | - | VL (*L. infantum*) |
| BR132017028144/ BR102012032499 | ELISA | K39-based | Modified peptide (tandem repeats) | Yes (H, C) | Yes (4) | - | VL (*L. infantum*) |
| WO201655836 | ELISA | K39-based | Modified peptide (tandem repeats) | Yes (C) | Yes (4) | - | VL (*L. infantum*) |
| BR102018017162 | ELISA | K39 | Recombinant protein expressed in plants | Yes (C) | No (n/a) | - | VL (*L. infantum*) |
| EP2756850 | ELISA | Kinesin-related protein KLO8 (K39-homologue) | Recombinant peptide | Yes (H, C) | Yes (4) | asympt; HIV + | VL (*L. donovani*); PKDL |
| BR102014013195/ BR102015012622 | ELISA | MAPK | Epitope | Yes (H, C) | Yes (4) | - | VL (*L. infantum*); TL (*L. braziliensis*) |
| BR102015012623/ BR102014013193 | ELISA | MAPK3 | Epitope | Yes (H, C) | Yes (4) | - | VL (*L. infantum*); TL (*L. braziliensis*) |
| BR102012032022/ BR102/WO201491463 | ELISA | HSP 83–1, MAPK and MAPK3 | Recombinant proteins | Yes (C) | Yes (4) | - | VL (*L. donovani, L. infantum*); TL (*L. braziliensis, L. major, L. mexicana*) |
| IN2010DE02939 | ELISA | BHU Pl, BHU P2, BHU P3 | Gel-digested peptides | Yes (H) | Yes (4) | - | VL (*L. donovani*) |
| WO2020168402 | ELISA | Lci2, Lci3 and Lci 12 | Chimera | Yes (H, C) | Yes (4) | HIV+ | VL (*L. infantum*) |
| BR102015032498 | ELISA | Linj.11.0370 | Protein and epitopes | Yes (H) | Yes (4) | Sub-genus specific | VL (*L. infantum, L. donovani*); TL (*L. major, L. mexicana, L. amazonensis*) |
| BR102014028172 | ELISA | r-cathepsin | Protein and epitope | Yes (H, C) | Yes (4) | - | VL (*L. infantum*); TL (*L. amazonensis*) |
| WO2011153602 | ELISA | GDPase | Recombinant protein | Yes (C) | Yes (4) | - | VL (*L. infantum*); TL (*L. braziliensis, L. major*) |
| BR102018073191/ BR102019023354 | ELISA | Prohibitin | Recombinant protein | Yes (H, C) | Yes (4) | asympt | VL (*L. infantum*) |

(*Continued*)

**Table 3.** (Continued)

| Patent # | Assay type | Target | Antigen details | Test with clinical sample | Accuracy results/ Level of evidence* | Test details | *Leishmania* species tested |
|---|---|---|---|---|---|---|---|
| BR102017005135 | ELISA | Putative GTP binding protein (Myxo) | Recombinant protein | Yes (C) | Yes (4) | - | VL (*L. infantum*) |
| EP3017305 | ELISA | H2A Histone, ARP | Epitopes | Yes (H) | Yes (4) | - | TL (*Leishmania* sp.) |
| BR102018067827 | ELISA | MPP | Recombinant protein | Yes (C) | No (n/a) | - | VL (*L. infantum*) |
| WO202041849 | ELISA | LPG | Purified from parasite culture | Yes (H) | Yes (4) | asympt | VL (*L. infantum*) |
| BR102018067309 | ELISA | LPG3 | Recombinant protein | Yes (C) | Yes (4) | - | VL (*L. infantum*) |
| BR102020008460 | ELISA | LiHyJ | Recombinant protein and epitope | Yes (H, C) | Yes (4) | asympt | VL (*L. infantum*) |
| BR 102020025878 | ELISA | LiHyC, LiHyG | Recombinant proteins and epitope | Yes (H, C) | Yes (4) | asympt | VL (*L. infantum*) |
| US20130236484 | ELISA | Isd, Txn1, Ntf2 | Peptides purified from patient urine (alone or combined) | Yes (H) | Yes (4) | Antigen detection in urine | VL |
| BR102014031331 | ELISA | Alpha-tubulin HSP83.1, HSP70 and K39 | Chimera | Yes (C) | No (n/a) | asympt | VL (*L. donovani*) |
| BR102015016162 | ELISA | Alpha-tubulin, HSP83.1, HSP70, K39, haspb2 | Chimera | Yes (C) | Yes (4) | - | VL (*L. infantum*) |
| BR102019004212 | ELISA | Alpha-tubulin, HSP83.1, HSP70, K39, haspb2 | Chimera | Yes (H, C) | No (n/a) | asympt | VL (*L. infantum*) |
| BR102020015591 | ELISA | Sequence only | Chimera | Yes (H) | Yes (4) | - | TL (*L. braziliensis*) |
| BR201105461 /BR132013001271/ WO201219268 | ELISA | Several[1+] | Gel-digested peptides | Yes (C) | Yes (4) | asympt | VL (*L. infantum*) |
| BR102016005090 | ELISA | Several[2+] | Gel-digested peptides | Yes (H) | No (n/a) | - | TL (*L. braziliensis, L. amazonensis*) |
| WO201811738 | ELISA | Several[3+] | Gel-digested peptides | Yes (H) | Yes (4) | asympt | VL (*L. infantum*) |
| EP3373950 | ELISA | Several[4+] | Peptide combination | Yes (H) | No (n/a) | - | VL |
| WO201597654 | ELISA | Sequence only | Epitopes | Yes (C) | Yes (4) | asympt | VL (*L. infantum*); TL (*L. braziliensis*) |
| BR102018016009 | ELISA | SMP-3 | Recombinant protein | Yes (H, C) | Yes (4) | asympt | VL (*L. infantum*); TL (*L. braziliensis*) |
| BR102013013069 | ELISA | Sequence only | Epitopes | Yes (C) | No (n/a) | asympt | VL (*L. infantum*) |
| WO2017103909 | ELISA | Sequence only | Epitopes | Yes (H) | Yes (4) | - | CL (*L. braziliensis*) |
| BR102012033552 | ELISA | Sequence only | Epitopes (isolated, in combination, or polymerized) | Yes (H) | No (n/a) | - | VL (*L. infantum*) |
| BR102014004107 | ELISA | Sequence only | Epitope | Yes (H, C) | Yes (4) | - | VL (*L. infantum*); TL (*L. braziliensis*) |
| BR102017013604 | ELISA | Sequence only | Epitopes | Yes (H, C) | Yes (4) | - | VL (*L. infantum*); TL (*L. braziliensis*) |
| WO2017109763 | ELISA | Sequence only | Epitopes | Yes (H) | Yes (4) | - | MCL (*L. braziliensis*) |
| BR102015017724 | ELISA | Sequence only | Epitopes (isolated or combination) | Yes (H) | Yes (4) | - | TL (*L. braziliensis*) |
| WO2018109753 | ELISA | Sequence only | Epitope | Yes (C) | Yes (4) | asympt | VL (*L. infantum*) |
| WO201641040 | ELISA | Lc36 | Recombinant peptide | Yes (C) | Yes (4) | - | VL (*L. infantum*) |
| CN102590508 | ICT | SLA | Soluble antigens | No | No (n/a) | - | VL |
| CN104142400 | ICT | SLA | - | No | No (n/a) | Antibody-based test | VL |

(*Continued*)

**Table 3.** (Continued)

| Patent # | Assay type | Target | Antigen details | Test with clinical sample | Accuracy results/ Level of evidence* | Test details | *Leishmania* species tested |
|---|---|---|---|---|---|---|---|
| BR102019020805 | Biosensor | - | Aptamer | No | No (n/a) | - | VL, TL |
| BR102019014136 | Biosensor | PLA2 | n/a | No | No (n/a) | Antibody-based test | TL (*L. amazonensis*) |
| KR20120024290 | LA | SLA | Soluble antigens | Yes (H) | No (n/a) | - | VL (*L. infantum*) |
| BR102017006706 | Flow Cytometry | Lci1A e Lci2B | Recombinant proteins | Yes (C) | Yes (4) | asympt | VL (*L. infantum*) |
| BR102012004742 | Flow Cytometry | *Leishmania* whole extract | *Leishmania* extract | Yes (C) | Yes (4) | - | VL (*L. infantum*) |
| BR102012005567 | Flow Cytometry | *Leishmania* whole extract | - | Yes (H) | Yes (4) | - | VL (*L. infantum*); TL (*L. amazonensis, L. braziliensis*) |
| EP3190412 | Skin Test | SLA, Prx | Soluble antigens and Recombinant protein | No | No (n/a) | - | CL (*L. major*) |

**Codes:** †Identified by Blast; asympt—detects asymptomatic infection; n/a—information not available; + Listed in S3 Text; Ind inventor—independent inventor. **Test codes**: CLIP—Crosslink immunoprecipitation; ICT- Immunochromatographic test; LA–Latex Agglutination Assay. **Gene codes:** LAg—Promastigote non-recombinant membrane antigen; SLA—Soluble *Leishmania* antigen; LACK—*Leishmania*-activated C-kinase antigen; MAPK—MAP kinase; HSP—Heat shock protein; NH—Nucleoside hydrolase; HbR—Hemoglobin receptor; Isd—Iron superoxide dismutase; Txn1—Tryparedoxin; EF-2—Elongation factor 2; ARP—Acidic ribosomal protein; LPG—Lipophosphoglycan; Isd—Iron superoxide dismutase; Txn -Tryparedoxin; Ntf2—Nuclear transport factor 2; Haspb—Hydrophilic acylated surface protein b; PLA2—Phospholipase A2; Prx—Mitochondrial peroxiredoxin; MPP—Metallo-peptidase, Clan ME, Family M16. **Antigen classification:** Recombinant (the antigen itself is identical to natural form); modified (recombinant and different from natural form); Chimera; Epitope (may also test the whole protein), Epitope combination (separate epitopes used in combination—may also test the whole proteins); *Level of evidence based on Turlik, 2009 [19]

issue given the limited resources in affected regions and the need for continuous surveillance to control and eradicate the disease. For epidemiological surveillance of canine visceral leishmaniasis, testing of a large number of animals in a short period of time with an acceptable precision is essential [33]. Tests that could satisfy these criteria include rapid antibody detection tests (with greater potential for VL), antigen detection tests (for VL and TL) and isothermal molecular methods (such as LAMP and RPA, for LV and LT). However, the experimental validation contained in our landscape documents is insufficient to make a recommendation as to which of the disclosed tests with these characteristics are more likely to pass clinical validation.

Considering the low levels of jointly owned patents, there is a need to strengthen collaborative networks focused on developing diagnostic tests for leishmaniasis. Partnerships between different players in the innovation process, such as governmental agencies, international organizations, academic and private sectors, can be used to maximize the strengths of each partner and combine the expertise in the technical field with know-how to develop a product that meets market needs. Such partnerships include, but are not limited to, product development partnerships (PDPs), open innovation, public–private partnerships (PPPs), joint ownership of

**Table 4. Other methods.**

| Patent number | Assay type | Target | Antigen details | Test with clinical sample | Accuracy results/ Level of evidence* | Test details | *Leishmania* species tested |
|---|---|---|---|---|---|---|---|
| ES2727968 | GC/ Q-TOF | - | Aptamer | No | No (n/a) | Detection of volatile organic compound in an exhalate sample | CL |

**Codes:** GC/Q-TOF (Gas chromatography with quadrupole time-of-flight mass spectrometry).

laboratories, among others. FIND is a successful example of a not-for-profit organization using a PPP business model for the development and implementation of diagnostic tools for poverty-related diseases. It has developed 24 diagnostic tools since 2003, including a LAMP based diagnostic test for VL. Their work on marginalized populations focuses on developing a diagnostic pipeline aligned with needs identified in the WHO's 2021–2030 roadmap to contribute to the elimination of NTDs [34].

Indeed, the experience gained in the globe response to NTDs suggests that, in addition to multisectoral efforts, international multilateral coordination is crucial: the Fifth Progress Report on the London Declaration on NTDs accredits progress made in the global combat of NTDs to (i) intricate public-private partnerships, involving coordination with non-governmental organizations, industry, donors, academic institutions, endemic country governments and front-line health workers and (ii) national and regional ownership, i.e., the translation of international targets into national goals and strategies, with the support of the international community [35].

Given the low level of contribution of the private sector to patent filings and the fact that less than 1% of R&D funding comes from "industry" (pharmaceutical companies and biotechnology firms) [36], incentives should be given to enhance the participation of the corporate sector. One possible strategy is to use push-pull mechanisms to reduce R&D costs and increase market attractiveness, as already used for NTD drug development. These include targeted R&D tax credits (direct governmental contribution to companies, designed to promote R&D in specific areas), rewards and prizes awarded for the development of products that meet specific requirements, advance purchase commitments, open source models that encourage collaboration and resource sharing between the private sector and academia, support for requirements needed for regulatory approval and mechanisms to fast track analyses by regulatory agencies [6, 37, 38]. A more integrated approach focusing on multiple NTDs diagnostic platforms could also reduce the market failure inherent to NTDs. Such an approach is suggested by WHO's 2021–2030 roadmap [4] and a strategic framework for the integrated control and management of skin NTDs, which includes CL and PKDL, was specifically launched in 2022 [39]. In fact, the need for a multiplex platform for skin NTD diagnosis is on DTAG's agenda, the requirements of which are still to be defined [40, 41]. At last, defining clear processes for test validation and adoption of tests by programs are also needed to improve the rate at which new tests can be introduced into public health programs.

Judging by the experimental evidence contained in patent documents, incentives are needed to (i) stimulate new inventions in the field of leishmaniasis diagnosis, (ii) align such inventions with market needs and (iii) help push existing inventions beyond the pre-clinical phase. These incentives include increased funding. Global leishmaniasis diagnosis research funding between 2007–2020 amounted to 33 million dollars, according to the G-finder data portal. This accounts for less than 5% of total R&D funding for the disease ($730 million) [36]. In view of the findings of the current patent landscape, it seems that not only more funds must be invested, but more funds allocated for the development of tests that meet market needs. A recent manifest by the Network of Researchers and Collaborators in Leishmaniasis (RedeLeish) recognizes the need for funding and calls attention to the fact that CL funding constraints are even more severe [42].

Finally, we must emphasize that patent landscapes reflect the current patent situation in each field, and do not consider ensuing experimental evidence obtained after patent application, unless they have been included in subsequent patent documents. Therefore, as with all patent landscape analyses, our results must be interpreted with caution. Overall, our results indicate that from a public policy perspective the development of diagnostic tests for leishmaniasis needs leveraging, as most tests revealed in the patent documents do not fulfill the critical

gaps for disease control mentioned in the WHO roadmaps for NTDs. Although these results may be discouraging, we should acknowledge that recent developments on diagnostic methods in general, including rapid and low-cost approaches, offer a positive prospect for the development of new tools to address public health needs.

## Supporting information

**S1 Data. Dataset compilation.**
(DOCX)

**S1 Text. Specific search strings used to gather our dataset.**
(DOCX)

**S2 Text. List of patents that were manually grouped.**
(DOCX)

**S3 Text. Full description of proteins listed on Table 3 under the heading "several".**
(DOCX)

## Acknowledgments

The authors would like to thank Dr Fabio Zicker for the corrections and constructive insights.

## Author Contributions

**Conceptualization:** Alice Fusaro Faioli.

**Data curation:** Daniel Moreira de Avelar, Camila Chaves Santos, Alice Fusaro Faioli.

**Formal analysis:** Daniel Moreira de Avelar, Camila Chaves Santos, Alice Fusaro Faioli.

**Investigation:** Alice Fusaro Faioli.

**Methodology:** Daniel Moreira de Avelar, Alice Fusaro Faioli.

**Project administration:** Alice Fusaro Faioli.

**Supervision:** Alice Fusaro Faioli.

**Validation:** Alice Fusaro Faioli.

**Visualization:** Alice Fusaro Faioli.

**Writing – original draft:** Daniel Moreira de Avelar, Alice Fusaro Faioli.

**Writing – review & editing:** Daniel Moreira de Avelar, Camila Chaves Santos, Alice Fusaro Faioli.

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
