## [Decision Letter · Decision Letter 0]

2 Aug 2023

PGPH-D-23-00388

Developments in Leishmaniasis diagnosis: a patent landscape from 2010 to 2022

Dear Dr. Fusaro Faioli,

Thank you for submitting your manuscript to PLOS Global Public Health. After careful consideration, we feel that it has merit but does not fully meet PLOS Global Public Health’s publication criteria as it currently stands. Therefore, we invite you to submit a revised version of the manuscript that addresses the points raised during the review process.

The manuscript has been evaluated by tworeviewers, and their comments are available below. They have provided a number of suggestions to improve your manuscript. Could you please revise the manuscript to carefully address the concerns raised?

We look forward to receiving your revised manuscript.

Kind regards,

Marianne Clemence

Staff Editor

Journal Requirements:

1. We have noticed that you have uploaded Supporting Information files, but you have not included a list of legends. Please add a full list of legends for your Supporting Information files after the references list. 

Additional Editor Comments (if provided):

Reviewers' comments:

Reviewer's Responses to Questions

**Comments to the Author**

1. Does this manuscript meet PLOS Global Public Health’s publication criteria? Is the manuscript technically sound, and do the data support the conclusions? The manuscript must describe methodologically and ethically rigorous research with conclusions that are appropriately drawn based on the data presented.

Reviewer #1: Yes

Reviewer #2: Yes

2. Has the statistical analysis been performed appropriately and rigorously?

Reviewer #1: Yes

Reviewer #2: Yes

3. Have the authors made all data underlying the findings in their manuscript fully available (please refer to the Data Availability Statement at the start of the manuscript PDF file)?

Reviewer #1: Yes

Reviewer #2: Yes

4. Is the manuscript presented in an intelligible fashion and written in standard English?

Reviewer #1: Yes

Reviewer #2: Yes

5. Review Comments to the Author

Reviewer #1: The manuscript “Developments in Leishmaniasis diagnosis: a patent landscape from 2010 to 2020” describes the analysis of patents filed for Leishmaniasis diagnostics and progress toward filling gaps identified by WHO for diagnosis. The authors describe molecular and serological diagnostic tests that have been experimentally validated in patent filing. The authors discuss several areas that can improve diagnostic research and innovation incentives. The paper is thorough, clearly written, the limitations are clearly stated, and the analysis is consistent with the conclusions. This will be of value to the scientific community to understand the current state of leishmaniasis diagnostics and promote innovation strategy. Minor comments for improvement are below:

1. Line 83: “results” should replace “resulting”

2. Check spacing consistency between words and sentences.

3. Line 412: rK39, should K be capitalized? Small case is used throughout, k39

4. Line 481: needs a period after the citation bracket

Reviewer #2: The present manuscript assesses the recent patent landscape for diagnostic tests for leishmaniasis. This is an interesting exercise in the current scenario of actions proposed to control neglected tropical diseases (NTDs), in this case leishmaniasis, aligned with Sustainable Developing Goal 3 (SDG3) and the World Health Organization (WHO) Road Map for NTDs.

The findings of this exercise highlight the need of increase collaborations and investments to develop much needed diagnostic tests for leishmaniasis, which will help addressing goals in SDG3 and WHO Road Map for NTDs.

The process followed by the authors to assess this landscape is clear and conclusions and recommendations reflect the findings.

This work should encourage and guide the scientific community in the development of tests to address diagnostic gaps for leishmaniasis.

There are only minor points that I would like to highlight for consideration by the authors:

1. Review the text for typos or missing words, examples:

1a. Page 1 Abstract "...revealed are consistent WITH the needs..."

1b. Page 8, line 86. "...published IN the scientific litearature..."

1c. Page 8, line 93. "...consistent WITH the needs..."

2. Page 7, line 78. PLease note that the target product profile mentioned is for dermal leishmaniasis, not only for cutaneous leishmaniasis (CL).

3. It may worth mentioning the development of target product profiles for diagnostic tests for visceral leishmaniasis https://www.who.int/news-room/articles-detail/call-for-public-consultation-----target-product-profiles-(tpp)-for-visceral-leishmaniasis-diagnostics

4. It may worth mentioning work done by the WHO Diagnostic Technical Advisory Group for NTDs https://www.who.int/groups/diagnostic-technical-advisory-group-for-neglected-tropical-diseases/about-us#:~:text=WHO%20has%20established%20a%20Diagnostic,process%20is%20able%20to%20properly

And the report of their different meetings, where priorities for leishmaniasis diagnostic tests are discussed, example: https://www.who.int/publications/i/item/9789240003590

Reports on up to the fifth consultative meeting can be found in who web site.

6. PLOS authors have the option to publish the peer review history of their article (what does this mean?). If published, this will include your full peer review and any attached files.

**Do you want your identity to be public for this peer review?** For information about this choice, including consent withdrawal, please see our Privacy Policy.

Reviewer #1: No

Reviewer #2: No

---

## [Editor Report · Decision Letter 1]

3 Oct 2023

PGPH-D-23-00388R1

Developments in Leishmaniasis diagnosis: a patent landscape from 2010 to 2022

Dear Dr. Fusaro Faioli,

Thank you for submitting your manuscript to PLOS Global Public Health. After careful consideration, we feel that it has merit but does not fully meet PLOS Global Public Health’s publication criteria as it currently stands. Therefore, we invite you to submit a revised version of the manuscript that addresses the points raised during the review process.

There are points which are to be carefully revised:

1. Review the text for typos / missing words/ spacing consistency  etc.

2. rk39 test should be corrected to  rK39 and maintain consistency in all instances

3. Page 7, line 78. Please note that the target product profile mentioned is for post kala azar dermal Leishmaniasis (PKDL) and also for cutaneous leishmaniasis (CL)

4. It may be worth mentioning work done by the WHO Diagnostic Technical Advisory Group for NTDs https://www.who.int/groups/diagnostic-technical-advisory-group-for-neglected-tropical-diseases/about-

This manuscript assess the recent patent landscape for diagnostic tests for Leishmaniasis. This manuscript is important in the present scenario of the elimination/control of neglected tropical disease and also to assess the gaps described in the  WHO NTD Roadmap 2021-2030 regarding elimination of Leishmaniasis.. The authors deserve mention for the discussion on different areas to improve the diagnostic research and innovations. This helps in understanding the various diagnostic tests and to promote innovation strategies by the scientific communit

Please submit your revised manuscript by  7 days, If you will need more time than this to complete your revisions, please reply to this message or contact the journal office at globalpubhealth@plos.org. Please include the following items when submitting your revised manuscript:

We look forward to receiving your revised manuscript.

Kind regards,

Suma Thankamma Krishnasastry, MBBS, MD,DNB

Academic Editor
---

## [Editor Report · Decision Letter 2]

10 Oct 2023

Developments in Leishmaniasis diagnosis: a patent landscape from 2010 to 2022

PGPH-D-23-00388R2

Dear Dr Fusaro Faioli,

We are pleased to inform you that your manuscript 'Developments in Leishmaniasis diagnosis: a patent landscape from 2010 to 2022' has been provisionally accepted for publication in PLOS Global Public Health.

Best regards,

Suma Thankamma Krishnasastry, MBBS, MD,DNB

Academic Editor